# Spatial autocorrelation and epidemiological survey of visceral leishmaniasis in an endemic area of Azerbaijan region, the northwest of Iran

Davoud Adham[1], Eslam Moradi-Asl[1]*, Abbasali Dorosti[2], Simin Khaiatzadeh[2]

1 Department of Public Health, School of Public Health, Ardabil University of Medical Sciences, Ardabil, Iran,
2 CDC, Health Center of Tabriz, Tabriz University of Medical Sciences, Tabriz, East Azerbaijan, Iran

* moradiasl83@yahoo.com

**Data Availability Statement:** All relevant data are within the manuscript and its Supporting Information files.

## Abstract

Visceral leishmaniasis (VL) is a common infectious disease that is endemic in Iran. This study aimed to investigate the spatial autocorrelation of VL in the northwest of Iran. In this cross-sectional study, the data of all patients were collected in 2009–2017 and analyzed by SPSS23 and Moran's and General G Index. The MaxEnt3.3.3 software was used to determine the ecological niche. A big hot spot area was identified in five counties in the northwest of Iran. More than 70% of the cases were reported from these regions, and the incidence rate increased in the northwest of Iran from 2013 to 2017. Seasonal rainfall and average daily temperature were the most important climate variables affecting the incidence of VL in this region (p < 0.05). Therefore, it can be concluded that VL in the northwest of Iran is expanding to new areas along the border with the Republic of Azerbaijan, and the northeastern section of this region is a high-risk area.

## Introduction

Arthropod-borne diseases are one of the most important health problems in the world. Today, more than one-third of infections are caused by communicable diseases by vectors. Leishmaniasis is a vector-borne disease transferred by more than 20 types of *Leishmania* parasites belonging to the *Kinetoplastida* class and the family *Trypanosomatidae* [1, 2]. Leishmaniasis has been reported in more than 101 countries [3], and over 350 million people face the grave risk of the disease [4]. The most important vectors of the leishmaniasis in the old world are sandflies of the genus *Phlebotomus* and *Lutzomyia* in the new world [5]. In terms of clinical symptoms, leishmaniasis is classified into the following varieties: cutaneous leishmaniasis (CL), visceral leishmaniasis (VL) and mucocutaneous leishmaniasis (MCL) [6, 7]. VL, also known as kala-azar, is an intense, hard form of leishmaniasis [8]. It is estimated to have caused 0.2–0.4 million cases worldwide resulting in over 40,000 deaths annually [9, 10].

VL in Iran is of the Mediterranean type and is caused by *Leishmania infantum* [7, 11]. Dogs and canines [12, 13] have been introduced as the main reservoirs, and 100–300 endemic and

**Funding:** This study was funded Ardabil University of Medical Sciences.

**Competing interests:** The authors have declared that no competing interests exist.

sporadic human cases annually occur in different regions in Iran [14]. After the observance of the first case of VL in Iran in 1949, at least four main foci of the disease have been investigated thus far in some parts of Ardabil, Fars, East Azerbaijan and Bushehr provinces where the most important focus of the disease has been reported to be the northwestern region of Iran (Ardabil and East Azerbaijan) [14, 15]. In Iran, infections occur more in children, and more than 89% of the patients in endemic areas are children under the age of five years [16]. The most important symptoms of VL include fever, hepatosplenomegaly, and anemia [17, 18]. Currently, 25–50% of VL cases occur in the northwest of Iran and hence it is one of the most important centers of the disease in the country [19, 20].

The use of Geographic Information System (GIS) is a new approach in the study of vector-borne diseases (VBDs), such as VL, and has led to significant changes in data interpretation and decision making [21]. Using geographic technology, researchers now have the opportunity to get informed about the distribution of the disease and its extent, identify and control the high-risk areas of the disease and take the necessary environmental interventions. Accordingly, there is a high correlation between the occurrence of the VL disease and the environmental factors involved in it in terms of geographical distribution and their relationship with the disease [22]. In an 18-year study that examined the status of VL in Italy based on GIS and the MaxEnt software, the overall increase in the disease was found to be related to the level of the patients' immune system and the impact of the social status of the endemic region on individuals where the disease was sporadic [23]. To understand and predict where the outbreak of diseases can happen, the ecological niche models and drawings of different maps can be used with the MaxEnt and GIS software, respectively [24]. Ecological niche modeling is also used to understand the probability of the presence of a disease in a specific location [25]. In fact, these models help find answers to ecological questions and identify the distribution of diseases that can be potentially epidemic disease [26].

Accordingly, this study aimed to investigate the geographical distribution of VL in three provinces of Ardabil, East Azerbaijan and West Azerbaijan (northwestern Iran) and determine the high-risk areas of the disease in the provinces and its relation with environmental and geographical factors. Another objective of the study was to determine the environmental suitability for VL in northwestern Iran for prevention and control of diseases.

## Material and methods

### Study area

Azerbaijan is located in the northwest of Iran, at the intersection of Alborz, Zagros and Caucasus Mountains and is a mainly mountainous region consisting of Ardabil, East Azerbaijan and West Azerbaijan provinces. This area is between 34.550–38.4853˚ N and 45.0001–48. 911˚E (Fig 1). In most parts of the west and northwest of Azerbaijan, the weather is Mediterranean humid and in the southern region, it is Mediterranean with hot summers. Due to the European route, this region has obtained a special position. It borders the Republic of Azerbaijan, Armenia, and Turkey. According to the latest census in 2016, the population of the Azerbaijan region is 10.56% (8,445,291) of the total population of Iran.

### The ethics committee waived the need of consent from patients

**The implementation of the study.** In this retrospective cross-sectional study, the Data on VL cases were collected from Ardabil, East Azerbaijan and West Azerbaijan Provinces health centers during the last 9yr from January 2009 to the end of December 2017. The number of patients with Direct Agglutination Test (DAT) positive serologic tests was more than 1:3200 cases with medical records, diagnosed by physicians and treated in different parts of the

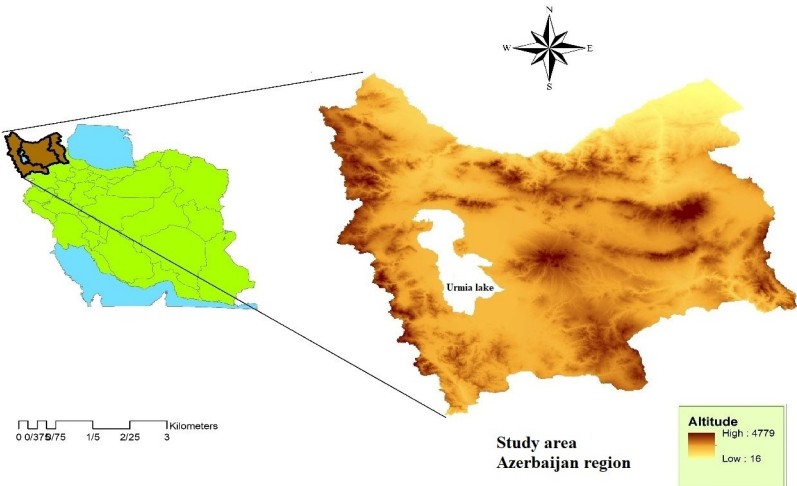

**Fig 1.**

province. By referring to health centers in different counties and reviewing patients' hospital and health records, demographic characteristics, including age, sex, place of residence, month and year of the disease as well as the results of the DAT test were collected. Then, the data collected were compared with the general statistics in the offices of the vice-chancellor for the health of the provinces of East Azerbaijan, West Azerbaijan, and Ardabil, and the results of the comparison were recorded. The collected data were stored in an ArcMap environment based on the patients' addresses.

Ethical grant was approved by Ardabil University of Medical Sciences by the following number: IR-ARUMS.REC.1397.112.

## Data analysis

The effect of variables, such as age, sex and location, on the prevalence (No. of VL cases/population size * 100000) of the disease was evaluated using SPSS version 23(Chicago, IL, USA), and T-test and Kruskal-Wallis statistical tests were run at a 95% confidence level. In order to perform the spatial analysis, the data were entered into the spatial data bank in the Arc-GIS10.4.1 software, and the distribution maps of the disease cases in different years were drawn in the ArcMap environment. Spatial autocorrelations and determination of high/low clustering of VL cases in different counties of the study area were estimated. The spatial auto-correlation tool in ArcGIS measures spatial autocorrelation based on both features of locations and values simultaneously. Given the VL cases data and the associated attribute (district border), the pattern of the disease (clustered, dispersed or random) was evaluated. Moran's I Index and General G value were measured, and both the score and p-values ($p < 0.05$) were calculated and used to evaluate the significance of the index [27]. Moran's I is a commonly used indicator of spatial autocorrelation. In this study, global Moran's I was used as the first measure of spatial autocorrelation. Its values range from−1 to 1. The value "1" means perfect positive spatial autocorrelation (high values or low values cluster together), while "−1" suggests perfect negative spatial auto-correlation (a checkerboard pattern), and "0" implies perfect spatial randomness [28, 29]. The High/Low Clustering (General G) tool measures how concentrated the high or low values are for a given study area. This tool calculates the High/Low General G value (observed & expected) & the associated Z score & p-value for a given input feature class [30].

Thus, the index was measured by the following formulas:

$$G = \frac{\sum_{i}^{N} = 1 \ \sum_{j}^{N} = 1 W_{ij} \ x_i \ x_j}{\sum_{i}^{N} = 1 \ \sum_{j}^{N} = 1 \ x_i \ x_j}, j \neq i \qquad I = \frac{N}{W} \frac{\sum i \ \sum j \ \omega_{ij}(x_i - x)(x_j - x)}{\sum i \ (x_i - x)}$$

## Modeling

To model the ecological niches of the parasite *L. infantum*, the coordinates of 42 points from different regions of Azerbaijan where more than two cases were reported in 2009–2017 were entered into the Excel environment and stored in the CSV format. Then, they were evaluated using the MaxEnt3.3 software [31]. A total of 19 climate variables were downloaded from the Worldclime website (www.worldclime.com) with a resolution of 30 sec (≅1 sq.km) and were used along with an elevation variable at the same resolution to evaluate and determine the appropriate ecological niches (Table 1). The Jackknife analysis was employed to find the most important variable in the model. Jackknife test was used to analyse the relationship between weather variables and distribution of *L.infantum* and the relevant variables were identified with percentages and non-relevant variables were assigned zero.

## Results

### Demography and the disease distribution

Over the 9-year period of the study, there were 202 cases with a positive result of VL disease throughout the northwest region of Iran, with the highest incidence in Ardabil Province (51%), and East Azerbaijan Province (47.50%) and the lowest in West Azerbaijan Province (1.50%). VL was distributed in 22 counties from all three provinces so that more than three cases were reported in 15 counties, most of them located in the western parts of Azerbaijan.

**Table 1. Variables used for MaxEnt modeling of VL distribution in Ardabil, East Azerbaijan and West Azerbaijan, Northwest of Iran.**

| Variable | Description | Contribution (%) |
|---|---|---|
| **Bio1** | Annual mean temperature (ºC) | 0 |
| **Bio2** | Mean diurnal range: mean of monthly (max temp–min temp) (˚C) | 21.6 |
| **Bio3** | Isothermality: (Bio2/Bio7) × 100 | 5.5 |
| **Bio4** | Temperature seasonality (SD × 100) | 5.9 |
| **Bio5** | Maximum temperature of warmest month (˚C) | 7.7 |
| **Bio6** | Minimum temperature of coldest month (˚C) | 3.4 |
| **Bio7** | Temperature annual range (Bio5 – Bio6) (˚C) | 0.1 |
| **Bio8** | Mean temperature of wettest quarter (˚C) | 0.1 |
| **Bio9** | Mean temperature of driest quarter (˚C) | 0 |
| **Bio10** | Mean temperature of warmest quarter (˚C) | 0 |
| **Bio11** | Mean temperature of coldest quarter (˚C) | 9 |
| **Bio12** | Annual precipitation (mm) | 0.1 |
| **Bio13** | Precipitation of wettest month (mm) | 4.5 |
| **Bio14** | Precipitation of driest month (mm) | 0 |
| **Bio15** | Precipitation seasonality (coefficient of variation) | 35.4 |
| **Bio16** | Precipitation of wettest quarter (mm) | 0.1 |
| **Bio17** | Precipitation of driest quarter (mm) | 3.8 |
| **Bio18** | Precipitation of warmest quarter (mm) | 0.2 |
| **Bio19** | Precipitation of coldest quarter (mm) | 2.1 |
| **Altitude** | Elevation from the sea level (m) | 0.7 |

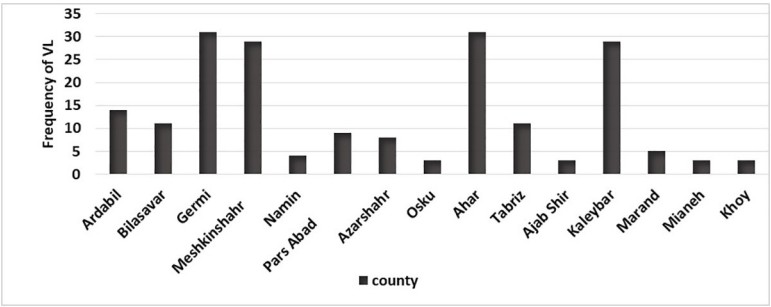

**Fig 2.**

Germi (17.80%), Ahar (14.90%), Meshkin Shahr (14.40%) and Kaleybar 10.40%) had the highest number of VL cases (Fig 2). This disease was reported from 119 areas in the Azerbaijan region of which 83% were rural areas and 17% were urban areas (Fig 3).

The monthly incidence of VL in the province showed that the highest infection rate was reported in March and February and the lowest in July and August. In other words, the seasonal occurrence of the disease occurred in winter (37.60%) and spring (30.70%) with a slight occurrence in summer (13.50%) (Fig 4). The highest number of VL cases recorded was 39 in 2009 and the fewest number of cases was eight in 2013. A total of 56% of patients were male and 44% were female. In terms of the age group, 98% of patients were under the age of 10 years, of which 68% were under two years of age and 32% were 2–10 years old; there was a statistically significant relationship between the ages of patients ($p < 0.05$). The relationship between the incidence rate of VL and the altitude showed that more than 60% of diseases occurred at altitudes of 1000 to 1500 m, and the results of the t-test revealed that this relationship was significant ($p = 0.005$). Besides, the results of the Kruskal-Wallis test showed that the incidence of VL was very high in 1150–1300 m altitude ($p = 0.017$) (Table 2).

## Spatial autocorrelation of VL in study area

The results of the Moran's Index and General G analysis indicated that the hot spot of the disease was located in the northeastern regions of Azerbaijan (Germi, Ahar, Meshkin Shahr and Kaleybar). Given the z-score of 5.6859, there is a less than 5% likelihood that this clustered pattern could be the result of random chance ($p = 0.000$). The high-low clustering report of General G factor given the z-score of 5.04132446743 showed that there is a less than 1% likelihood that this high-clustered pattern could be the result of random chance ($p = 0.000$). These

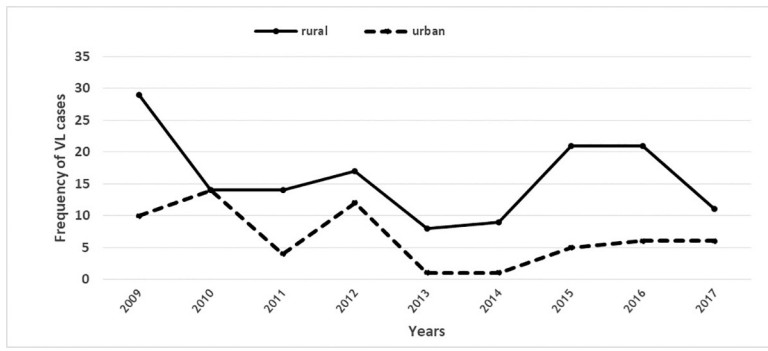

**Fig 3.**

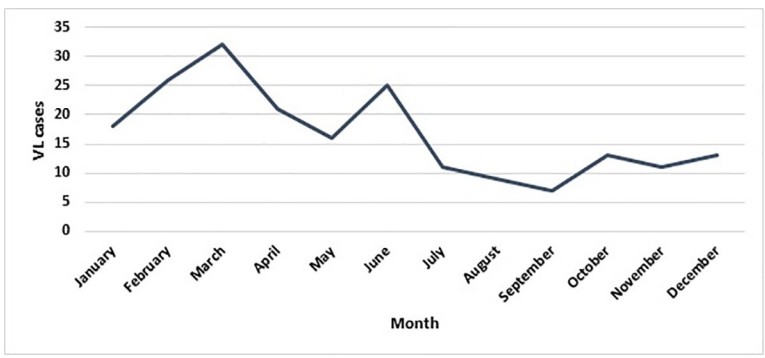

**Fig 4.**

infected areas share borders with the Republic of Azerbaijan; as a result, the risk of transmission of the disease in these areas is high (Figs 5–7) (Table 3).

## Modeling the ecologically suitable areas of VL

Based on the findings of this study, the northeastern parts of the Azerbaijan region on the common border between Iran and the Republic of Azerbaijan, including the counties of Germi, Meshkin Shahr, Ahar and Kaleybar, are among the high-risk areas for the transmission of disease and a suitable place for the growth of *L. infantum* parasite (Fig 6). The most important climate variables affecting the distribution of VL and *L. infantum* parasite in the Azerbaijan region were seasonal rainfall, average daily temperature range, the average temperature in cold seasons and maximum temperature in warm months. Once, the impact of each variable on the model was evaluated separately (Fig 8). In addition, the impact of all variables together on the model were precipitated again. The model showed that two factors; seasonality (35.40%) and mean diurnal range (21.60%) had the greatest impact on the occurrence of the disease (Table 1).

## Discussion

The investigation of the distribution of VL disease in the northwestern region of Iran in 2009–2017 indicated a large spread of the disease and an increase in the contaminated areas wherein nearly one-quarter of the region was infected. Throughout this 9-year period, 202 cases of the disease were identified and reported in the northwestern region of Iran. According to a previous study conducted in Iran in 1988, 5,244 cases of VL were detected of which 2,280 cases were observed in Ardabil Province, 2,020 cases in Fars Province and 175 cases in East Azerbaijan Province [32, 33]. More than 2,000 cases were diagnosed in 31 provinces of Iran in 2012 of which 44.60% were reported in the northwest of Iran [14], indicating that the northwestern region of Iran is one of the most important endemic foci of VL in Iran.

**Table 2. The results of T-test and Kruskal–walls test for incidence of VL in Ardabil, East Azerbaijan and West Azerbaijan, Northwest of Iran, 2009–2017.**

| T-test analysis | | | | Kruskal–walls test analysis | | | |
|---|---|---|---|---|---|---|---|
| Altitude (m) | Mean of incidence (*100000) | SD | P-value | Altitude (m) | Mean of incidence (*100000) | SD | P-value |
| 1000–1150 | 1276 | 134.21 | 0.005 | 1000–1150 | 371.25 | 532.34 | 0.017 |
| <1000 and >1150 | 867 | 618.65 | | 1150–1300 | 510.00 | 552.63 | |
| | | | | >1300 | 107.33 | 217.95 | |

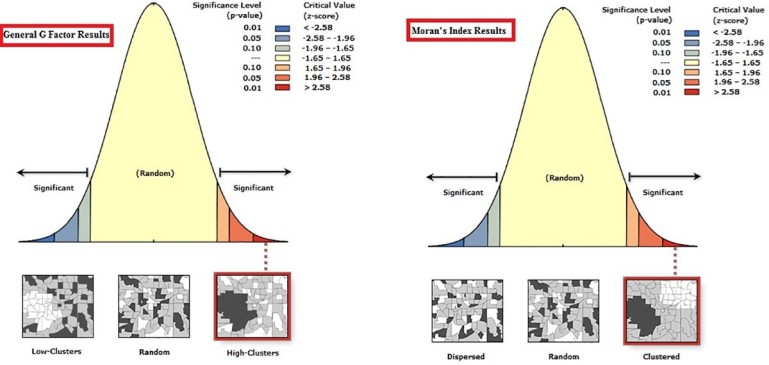

**Fig 5.**

The results of this study demonstrated that the disease has decreased in the northwest of Iran, but has been reported in more areas and parts than before. In another study carried out in 1996, three regions, Meshkin Shahr, Moghan and Ahar, were among the areas where the

**Fig 6.**

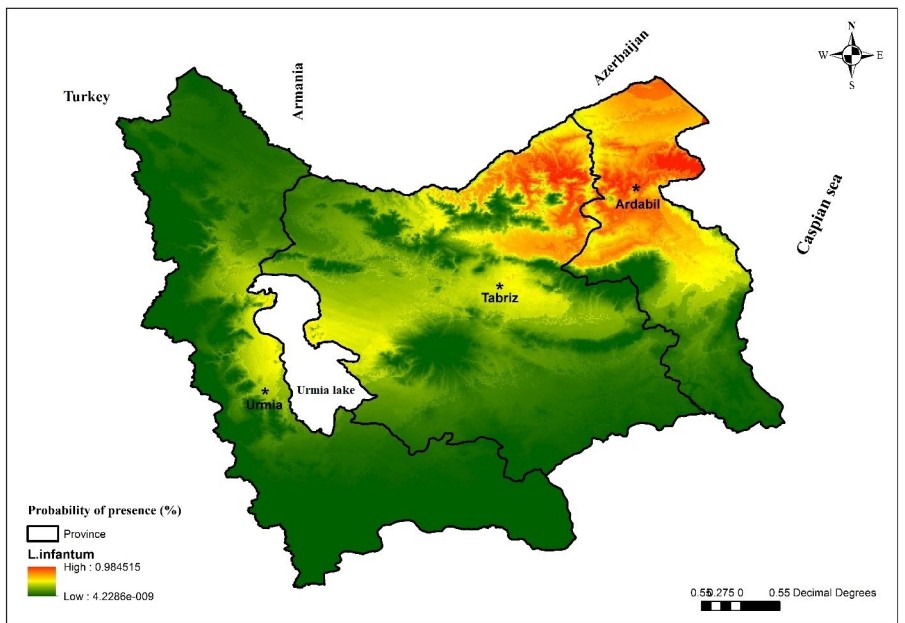

**Fig 7.**

disease was reported [34]. According to the results of this study, the disease cases were reported in 15 counties in the northwest so that the disease cases in seven counties were between 10 and 36, indicating that the distribution of the disease and the contaminated areas have generally increased. A study conducted before 2000 showed that many VL cases were reported from the three counties of Meshkin Shahr, Moghan and Ahar [26].

As observed in the study conducted by Marco in Brazil, the distribution of the disease and the number of contaminated areas have increased in different parts of the world. According to the findings of this study in Brazil, the incidence of VL reduced from 1999 to 2008, but its incidence spread to the various reservoirs and areas [35]. In India, the cases of VL dropped from around 76,500 cases in 1992 to around 12,000 cases in 2012, but the distribution of the disease increased in different states [36, 37]. Only in Afghanistan, the incidence of VL increased from 10,944 in 2003 to 32,145 cases in 2010; one of the main causes of this increase is the war in the country that has increased the number of diseases and areas of distribution [38].

The results of the study regarding patients' age indicated that 98% of patients were under the age of 10 and only 2% were above 10 years old. Due to the fact that VL in Iran is of Mediterranean type and its factor is *L. infantum* [39], as far as the age is concerned, this disease occurred at an early age, while previous research in Ardabil Province had shown that 17% of cases were under two years of age [40], now reaching 68%. In a study conducted in Ardabil

**Table 3. The results of hot-spot and autocorrelation analysis of VL in Ardabil, East Azerbaijan and West Azerbaijan, Northwest of Iran.**

| General G Summary | | Global Moran's I Summary | |
|---|---|---|---|
| **Observed General G:** | 0.000004 | **Moran's Index:** | 0.421890 |
| **Expected General G:** | 0.000001 | **Expected Index:** | -0.010101 |
| **Variance:** | 0.000000 | **Variance:** | 0.005772 |
| **z-score:** | 5.041324 | **z-score:** | 5.685973 |
| **p-value:** | 0.000000 | **p-value:** | 0.000000 |

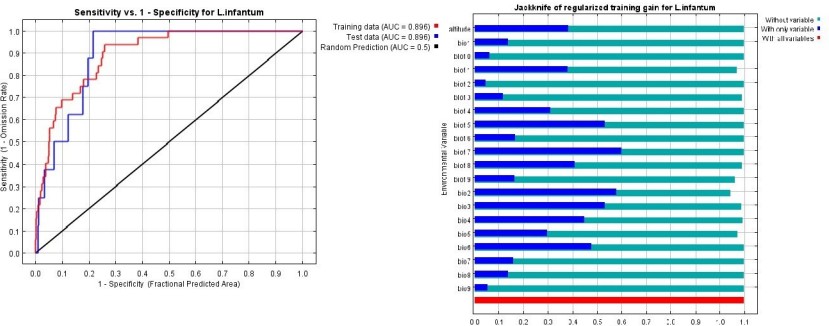

**Fig 8.**

Province in 2014 [16], 66% of cases in Ardabil Province were under two years old, while in 2017, this rate reached 70% [41].

In terms of gender, most cases were male (56%) ($p < 0.05$) indicating males are more likely to be infected. One of the reasons for such an observation can be the type of coverage for boys compared to girls. In other studies, in Pakistan and Brazil, the male/female sex ratio was 2/1, and the male patients were more than women [42–45].

More than 68% of cases occurred in winter and spring, and the incidence was low in summer and autumn. In terms of yearly incidence, the incidence of the disease declined by 2013, but from 2014 onward, there was a rising trend and according to the pattern of infectious diseases, it had a sine wave. In other areas of the world, including India, the cases of the disease have shown a sinusoidal pattern once every two or three years [36, 46]. The results of the analysis of Moran′s Index showed that the two counties of Germi and Meshkin Shahr in Ardabil Province and the two counties of Ahar and Kaleybar in East Azarbaijan Province had the highest risk of VL infection. These areas are among the high-risk areas with high transmission risk of infection in northwestern Iran. According to the results of spatial autocorrelation analysis tests, the distribution of VL was entirely clustered and not random. An important hot spot was located in the northeast of this area. These four regions share borders and are geographically interconnected. Considering that this area is one of the areas of Arasbaran forests and many people are engaged in agriculture and horticulture protecting their livestock and gardens by keeping dogs (the main reservoirs of the disease), they are more likely to develop the incidence of the disease than other areas. The main reservoirs of VL in Iran are dogs and Canidae, including jackals, foxes and wolves, and their abundance in the Arasbaran region is very high and the leishmaniasis parasite is often isolated and reported from the reservoirs in this region [12, 14, 15, 17, 36, 46].

The presence of definite vectors has caused the northwest region to be at risk for the transmission of the disease because out of six main vectors of VL in Iran, three main vectors are *Ph. kandelakii*, *Ph. perfiliewi* and *ph. tobbi* from the northwest region of Iran [47–51]; moreover, the abundance and diversity of sandflies species in this region are important. In a study conducted by Bavia et al. in the Bahia region of Brazil, using the GIS and RS, the high-risk areas of the disease were drawn by maps and the abundance of reservoirs and vectors were determined by matching; it was observed that the contamination of reservoirs and vectors was significantly higher in all areas where VL cases were present [52]. Likewise, in another study carried out in some parts of Brazil on the distribution and prediction of VL, using ecological niche model and climatic changes, Paulo Silva identified two new areas that were susceptible to VL located in the northern parts of the endemic areas and provided a map of high-risk areas [53].

Based on the results of the Jackknife test, two important seasonal factors, i.e. rainfall and temperature, had an important effect on the abundance of parasites and diseases in the

Azerbaijan region. Also, based on existing maps of areas with elevations between 1000 and 1500 meters above sea level, many cases of VL occurred in 1150–1300 m areas. The life cycle of the parasite in the body of the vector, including sandflies and the life cycle of the vectors, is also dependent on two important factors of temperature and humidity [54]; similarly, in a study conducted by Moradi-Asl et al. in Ardabil Province [41] and a study conducted by Han-afi-Bojd et al. in some regions of Iran [55], the effects of these two factors on the distribution of vectors and parasites and the incidence of disease were also emphasized.

The limitation of this study was that just the data from patients enrolled in the three provinces' health system centers were included in the study. The patients who were diagnosed and treated outside of the three provinces were unavailable. Another limitation was that we extracted data from health record documents and did not deal directly with patients.

## Conclusion

Based on the results of this study, the northwestern region of Iran is an important focal point for VL in Iran. Many programs have been undertaken to prevent and control the disease in the endemic areas of northwestern Iran, but most of the cases reported in the country are from the northwest of Iran (Ardabil Province and East Azerbaijan Province). In many counties, the new cases of the disease have been reported from new areas. This indicates the potential for the spread of the disease in these areas. Thus, it is necessary to carry out appropriate studies to investigate vectors in East Azerbaijan Province, evaluate the isolation of parasites from the vectors and determine their strains because the prevention and control of the disease, type of parasite and the type of sandflies transmitting the parasite should be determined in the field.

## Supporting information

**S1 Data.**
(XLSX)

**S2 Data.**
(XLS)

## Acknowledgments

The authors are grateful to all colleagues at the University of Medical Sciences and staff of health centers in all counties in Ardabil Province. We also like to thank Mr. D. Emdadi and Mr. J. Ebishvand.

## Author Contributions

**Data curation:** Davoud Adham, Eslam Moradi-Asl, Simin Khaiatzadeh.

**Formal analysis:** Eslam Moradi-Asl, Simin Khaiatzadeh.

**Investigation:** Abbasali Dorosti.

**Methodology:** Davoud Adham, Eslam Moradi-Asl.

**Writing – review & editing:** Eslam Moradi-Asl, Abbasali Dorosti.

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
