## [Decision Letter · Decision Letter 0]

27 Feb 2020

PONE-D-19-35837

Spatial Autocorrelation and Epidemiological Survey of Visceral Leishmaniosis in an endemic area of Azerbaijan Region, Northwest of Iran

PLOS ONE

Dear Dr Moradi-Asl,

Thank you for submitting your manuscript to PLOS ONE. After careful consideration, we feel that it has merit but does not fully meet PLOS ONE’s publication criteria as it currently stands. Therefore, we invite you to submit a revised version of the manuscript that addresses the points raised during the review process.

As pointed out by the two current reviewers of the manuscript, language needs extensive revision. Language strongly affects the quality of the review process. So, when the revised and language edited manuscript, with current reviewers’ 1 and 2 suggestions and comments incorporated, is resubmitted it will necessarily be distributed again for further review. While incorporating the current suggestions by reviewers 1 and 2, please include also the minor editorial suggestions made after this letter.

We would appreciate receiving your revised manuscript by Apr 12 2020 11:59PM. To enhance the reproducibility of your results, we recommend that if applicable you deposit your laboratory protocols in protocols.io, where a protocol can be assigned its own identifier (DOI) such that it can be cited independently in the future. For instructions see: http://journals.plos.org/plosone/s/submission-guidelines#loc-laboratory-protocols

We look forward to receiving your revised manuscript.

Kind regards,

Albert Schriefer, M.D., Ph.D.

Academic Editor

PLOS ONE

Additional Editor Comments (if provided):

- Improve titles in Results section to be more descriptive.

- Include legends to figures.

- Check figures axes titles. For example, Y axis in figure 4 is labeled ‘Vl’ instead of ‘VL’.

Journal Requirements:

2. In the ethics statement in the manuscript and in the online submission form, please provide additional information about the patient records used in your retrospective study. Specifically, please ensure that you have discussed whether all data were fully anonymized before you accessed them and/or whether the IRB or ethics committee waived the requirement for informed consent. If patients provided informed written consent to have data from their medical records used in research, please include this information.

3. Please ensure you have thoroughly discussed any potential limitations of this study within the Discussion section.

4. During your revisions, please note that a simple title correction is required to ensure correct spelling. The title should read: "Spatial Autocorrelation and Epidemiological Survey of Visceral Leishmaniasis in an endemic area of Azerbaijan Region, Northwest of Iran". Please ensure this is updated in the manuscript file and the online submission information.

"This work was supported by the Ardabil University of Medical Sciences (Project numbers 92218)."

"Ardabil University of Medical Sciences "

6. We note that you have indicated that data from this study are available upon request. PLOS only allows data to be available upon request if there are legal or ethical restrictions on sharing data publicly. For information on unacceptable data access restrictions, please see http://journals.plos.org/plosone/s/data-availability#loc-unacceptable-data-access-restrictions.

7. We note that Figures #1, 6 and 7 in your submission contain map images which may be copyrighted. All PLOS content is published under the Creative Commons Attribution License (CC BY 4.0), which means that the manuscript, images, and Supporting Information files will be freely available online, and any third party is permitted to access, download, copy, distribute, and use these materials in any way, even commercially, with proper attribution. For these reasons, we cannot publish previously copyrighted maps or satellite images created using proprietary data, such as Google software (Google Maps, Street View, and Earth). For more information, see our copyright guidelines: http://journals.plos.org/plosone/s/licenses-and-copyright.

1.    You may seek permission from the original copyright holder of Figures #1, 6 and 7 to publish the content specifically under the CC BY 4.0 license. 

8. Please include a copy of Table 2 which you refer to in your text on page 6 and Table 1 and 3 which you refer to in your text on page 7.

Reviewers' comments:

Reviewer's Responses to Questions

**Comments to the Author**

1. Is the manuscript technically sound, and do the data support the conclusions?

Reviewer #1: Partly

Reviewer #2: No

2. Has the statistical analysis been performed appropriately and rigorously? 

Reviewer #1: I Don't Know

Reviewer #2: No

3. Have the authors made all data underlying the findings in their manuscript fully available?

Reviewer #1: No

Reviewer #2: No

4. Is the manuscript presented in an intelligible fashion and written in standard English?

Reviewer #1: Yes

Reviewer #2: No

5. Review Comments to the Author

Reviewer #1: Dear Authors

Please consider the following comments:

1- Please یouble-check the spelling of the words in all manuscript. For example in title please change leishmaniosis to leishmaniasis.

2- The article needs to be edited in English by a native.

3- Please define in methodology how did you calculated the incidence of VL?

4- Please explain more about the method of analysis for Moran's Index and General G analysis. Did you used the points of disease or considered the counties as the analysis units.

5- Please cite a relevant reference for MaxEnt model.

6- The reference 24 cited in line 104 seems to be irrelevant to the analysis.

7- Fig 5 for the result of spatial auto-correlation should be changed with a map resulted from this analysis.

8- In all maps please use "Urmia Lake"

9- Table 1 was not included in the reviewed file.

10- Please discuss more about the role of weather and environmental variables on the habitat suitability for life cycle of VL.

Reviewer #2: This study analyzes the spatial patterns of visceral leishmaniasis in three provinces (Ardabil, East Azerbaijan and West Azerbaijan) in northwestern Iran from 2009 to 2017, as well as its relationship with environmental and geographical factors. The manuscript has a relevant theme, but needs an important review/organization in its content and writing.

-I suggest a comprehensive review of the English written in the manuscript (typographical / grammatical errors and scientific writing);

-Please describe clearly and comprehensively the main problem/object addressed in the manuscript;

-The introduction does not provide an entirely relevant description of the proposal and significance of the study. The problem is not significantly and concisely stated in the text;

-The methods needs to include some additional methodological details in the text (e.g., how morbidity data was extracted or made available; case definition; more information from the surveillance system responsible for the data, etc.), as well as a comprehensive and detailed description of all analyzes presented in the results;

-Overall, the results are not clearly presented and the interpretations and conclusions are not supported and justified by the results, but rather general statements. The limitations of this study are not presented in the text (e.g., coverage and quality of secondary data);

-The titles of figures and tables are not provided in the manuscript, and the tables were not included in the main file.

6. PLOS authors have the option to publish the peer review history of their article (what does this mean?). If published, this will include your full peer review and any attached files.

Reviewer #1: No

Reviewer #2: No

---

## [Author Response · Author response to Decision Letter 0]

13 May 2020

Dear editor of the Plos One

1) Thank you for your reply. I noted that our ethics committee waived the need for informed consent from the patients and the patients provided consent for the records to be used in this research.

 2.) Yes, the Supporting Information files named “GIS Data of VL.xlsx” and “VL Data.xls,” along with the data located in our manuscript, constitute the minimal data set required to replicate the conclusions in our study.

3.) Yes, “All relevant data are within the manuscript and its Supporting Information files.”

My dear editor, 

Thank you very much for that patiently and carefully to your comments and we will guide you. I hope I have answered your questions correctly. We worked hard on this study and took the time. This study was conducted in a very important area of endemic disease and with this study, high-risk areas are easily identified for VL disease control and prevention programs.

Best regards 

1) Thank you once again for your attention to our queries. You have stated that "This data not including the patients characters (name or address)." but it is unclear whether while reviewing patients’ hospital and health records any of the authors had accessed to potentially identifiable information or whether the authors have only accessed fully anonymized records. Could you please clarify this? Please also state in your ethics statement whether the ethics committee waived the need for consent from the patients or whether you obtained informed consent from the patients for their medical records to be used in research/this study. 

We after receiving the ethics code of the ethics committee of Ardabil University of Medical Sciences We used the database of health centers. This data includes patients who have been previously treated. We used demographic characteristics of patients and place of residence to study. The decision of the ethics committee will also be sent to the appendix.

2) Thank you for your additional data sharing information. If you would like to provide an Excel file containing your data, please upload your minimal anonymized data set to your PLOS ONE submission as a Supporting Information file.

Ok, we're sending two Excel files that include:

A. File the number of patients in the study area.

B. The incidence of VL files to different locations for GIS software is used.

My dear Anita Estes ,

PONE-D-19-35837R1

Spatial Autocorrelation and Epidemiological Survey of Visceral Leishmaniasis in an endemic area of Azerbaijan Region, Northwest of Iran

Thank you for better comments. 

1) Please include your tables as part of your main manuscript and remove the individual files. Please note that supplementary tables (should remain/ be uploaded) as separate "Supporting Information" files. ( Ok , was done.)

2) Thank you for removing the funding-related text from the Acknowledgments Section of your manuscript. Please let us know how you would like to update your Funding Statement. Currently, your Funding Statement reads as follows:

"Ardabil University of Medical Sciences"

(This study was funded Ardabil University of Medical Sciences)

3) Please amend your authorship list in your manuscript file to include all authors. (The names of the authors were checked and there was no problem).

4) We note that with regard to the copyright of Figures 1, 6, and 7, you wrote: "I dragged figures 1, 6 and 7 with the Arcmap GIS 10.4.1 software and uploaded the original file of GIS." (Sorry, it was a writing error. They were proper originally by GIS and we uploaded the GIS files . We did not dragged these figures from anywhere).

5) Please confirm whether the data uploaded as Supporting Information files constitutes the minimal data set, defined as the data set used to reach the conclusions drawn in the manuscript with related metadata and methods, and any additional data required to replicate the reported study findings in their entirety (https://journals.plos.org/plosone/s/data-availability#loc-minimal-data-set-definition). This may include: a.) The values behind the means, standard deviations and other measures reported; b.) The values used to build graphs; c.) The points extracted from images for analysis. (No, According to ethical approval, patient information is confidential and we can't share).

6) If there are ethical or legal restrictions to sharing third-party data used in your study, and the data included as Supporting Information files in your submission does not meet the requirements in the minimal data set definition above, please provide the following: a.) A description of the data set and the third-party source; b.) If applicable, verification of permission to use the data set; and c.) All necessary contact information others would need to apply to gain access to the data. Please note that it is not acceptable for an author to be the sole named individual responsible for ensuring data access. You can find more information on PLOS ONE’s policies regarding acceptable restrictions and third-party data via the following link: https://journals.plos.org/plosone/s/data-availability#loc-acceptable-data-access-restrictions. ( Yes, Based on patient information, we cannot share data.).

Thank you 

Dr.Eslam Moradi-Asl 

My dear editor and reviewers ,

I'm so sorry. We're in control of corona virus that's why it's late.

We tried to respond to all comments individually.We answered every question or comment in front of you.The English were Native.All shapes and maps were edited.We created all the maps ourselves with the Arcmap GIS 10.4.1 software.

We wish you all health and good time.

PONE-D-19-35837

Spatial Autocorrelation and Epidemiological Survey of Visceral Leishmaniosis in an endemic area of Azerbaijan Region, Northwest of Iran

PLOS ONE

Additional Editor Comments (if provided):

-Improve titles in Results section to be more descriptive. ( Ok , Edited ) 

- Include legends to figures. (Ok , Edited )

- Check figures axes titles. For example, Y axis in figure 4 is labeled ‘Vl’ instead of ‘VL’. Ok , Edited )

-. Please ensure you have thoroughly discussed any potential limitations of this study within the Discussion section. Ok, the limitations of the study were discussed at the end.

Journal Requirements:

(Ok, was done )

2. In the ethics statement in the manuscript and in the online submission form, please provide additional information about the patient records used in your retrospective study. Specifically, please ensure that you have discussed whether all data were fully anonymized before you accessed them and/or whether the IRB or ethics committee waived the requirement for informed consent. If patients provided informed written consent to have data from their medical records used in research, please include this information. (Ok, was done )

3. Please ensure you have thoroughly discussed any potential limitations of this study within the Discussion section. (Ok, was done )

4. During your revisions, please note that a simple title correction is required to ensure correct spelling. The title should read: "Spatial Autocorrelation and Epidemiological Survey of Visceral Leishmaniasis in an endemic area of Azerbaijan Region, Northwest of Iran". Please ensure this is updated in the manuscript file and the online submission information. (Ok, was done )

"This work was supported by the Ardabil University of Medical Sciences (Project numbers 92218)."

"Ardabil University of Medical Sciences " (Ok, was done )

6. We note that you have indicated that data from this study are available upon request. PLOS only allows data to be available upon request if there are legal or ethical restrictions on sharing data publicly. For information on unacceptable data access restrictions, please see http://journals.plos.org/plosone/s/data-availability#loc-unacceptable-data-access-restrictions.

The Excel and GIS format of the data and maps is sent to the attachment. 

7. We note that Figures #1, 6 and 7 in your submission contain map images which may be copyrighted. All PLOS content is published under the Creative Commons Attribution License (CC BY 4.0), which means that the manuscript, images, and Supporting Information files will be freely available online, and any third party is permitted to access, download, copy, distribute, and use these materials in any way, even commercially, with proper attribution. For these reasons, we cannot publish previously copyrighted maps or satellite images created using proprietary data, such as Google software (Google Maps, Street View, and Earth). For more information, see our copyright guidelines: http://journals.plos.org/plosone/s/licenses-and-copyright.

1. You may seek permission from the original copyright holder of Figures #1, 6 and 7 to publish the content specifically under the CC BY 4.0 license. 

*** I dragged figures 1, 6 and 7 with the Arcmap GIS 10.4.1 software and uploaded the original file of GIS.*****

8. Please include a copy of Table 2 which you refer to in your text on page 6 and Table 1 and 3 which you refer to in your text on page 7.

Ok , was done 

Reviewer #1: Dear Authors

Please consider the following comments:

1- Please double-check the spelling of the words in all manuscript. For example in title please change leishmaniosis to leishmaniasis. ( Ok , it is done)

2- The article needs to be edited in English by a native.( It was native).

3- Please define in methodology how did you calculated the incidence of VL?

R: (The effect of variables, such as age, sex and location, on the prevalence (No. of VL cases/population size * 100000) of the disease was evaluated using SPSS version 23(Chicago, IL, USA)

4- Please explain more about the method of analysis for Moran's Index and General G analysis. Did you used the points of disease or considered the counties as the analysis units.

R: (Moran's I Index and General G value were measured, and both the score and p-values (p < 0.05) were calculated and used to evaluate the significance of the index (27). Moran’s I is a commonly used indicator of spatial autocorrelation. In this study, global Moran’s I was used as the first measure of spatial autocorrelation. Its values range from−1 to 1. The value “1” means perfect positive spatial autocorrelation (high values or low values cluster together), while “−1” suggests perfect negative spatial auto-correlation (a checkerboard pattern), and “0” implies perfect spatial randomness (28, 29) . The High/Low Clustering (General G) tool measures how concentrated the high or low values are for a given study area. This tool calculates the High/Low General G value (observed & expected) & the associated Z score & p-value for a given input feature class(30).)

5- Please cite a relevant reference for MaxEnt model.( Ok , Reference 31 , Young N, Carter L, Evangelista P. A MaxEnt model v3. 3.3 e tutorial (ArcGIS v10). Fort Collins, Colorado. 2011.)

6- The reference 24 cited in line 104 seems to be irrelevant to the analysis.( Reference edited ) 

7- Fig 5 for the result of spatial auto-correlation should be changed with a map resulted from this analysis. (Figure 5 is the software output and is intended to illustrate the relationship).

8- In all maps please use "Urmia Lake" , ( Ok , Was done). 

9- Table 1 was not included in the reviewed file. ( This is for Editor) 

10- Please discuss more about the role of weather and environmental variables on the habitat suitability for life cycle of VL.

Reviewer #2: This study analyzes the spatial patterns of visceral leishmaniasis in three provinces (Ardabil, East Azerbaijan and West Azerbaijan) in northwestern Iran from 2009 to 2017, as well as its relationship with environmental and geographical factors. The manuscript has a relevant theme, but needs an important review/organization in its content and writing. ( Ok , The manuscript was edited native )

-I suggest a comprehensive review of the English written in the manuscript (typographical / grammatical errors and scientific writing); Ok , The manuscript was edited native

-Please describe clearly and comprehensively the main problem/object addressed in the manuscript; The introduction was edited 

-The introduction does not provide an entirely relevant description of the proposal and significance of the study. The problem is not significantly and concisely stated in the text; ( Ok , Accordingly, this study aimed to investigate the geographical distribution of VL in three provinces of Ardabil, East Azerbaijan and West Azerbaijan (northwestern Iran) and determine the high-risk areas of the disease in the provinces and its relation with environmental and geographical factors. Another objective of the study was to determine the environmental suitability for VL in northwestern Iran for prevention and control of diseases.)

-The methods needs to include some additional methodological details in the text (e.g., how morbidity data was extracted or made available; case definition; more information from the surveillance system responsible for the data, etc.), as well as a comprehensive and detailed description of all analyzes presented in the results; Ok, was edited ((In this retrospective cross-sectional study, the Data on VL cases were collected from Ardabil, East Azerbaijan and West Azerbaijan Provinces health centers during the last 9yr from January 2009 to the end of December 2017. The number of patients with Direct Agglutination Test (DAT) positive serologic tests was more than 1:3200 cases with medical records, diagnosed by physicians and treated in different parts of the province. By referring to health centers in different counties and reviewing patients’ hospital and health records, demographic characteristics, including age, sex, place of residence, month and year of the disease as well as the results of the DAT test were collected. Then, the data collected were compared with the general statistics in the offices of the vice-chancellor for the health of the provinces of East Azerbaijan, West Azerbaijan, and Ardabil, and the results of the comparison were recorded. The collected data were stored in an ArcMap environment based on the patients’ addresses.))

-Overall, the results are not clearly presented and the interpretations and conclusions are not supported and justified by the results, but rather general statements. The limitations of this study are not presented in the text (e.g., coverage and quality of secondary data); Once again the whole article was edited according to your comments and it was native.

-The titles of figures and tables are not provided in the manuscript, and the tables were not included in the main file.

---

## [Decision Letter · Decision Letter 1]

24 Jun 2020

PONE-D-19-35837R1

Spatial Autocorrelation and Epidemiological Survey of Visceral Leishmaniasis in an endemic area of Azerbaijan Region, Northwest of Iran

PLOS ONE

Dear Dr. Moradi-Asl,

Thank you for submitting your manuscript to PLOS ONE. After careful consideration, we feel that it has merit but does not fully meet PLOS ONE’s publication criteria as it currently stands. Therefore, we invite you to submit a revised version of the manuscript that addresses the points raised during the review process.

More specifically, please fully incorporate the two remaining comments made by reviewer 1 in the revised version of the manuscript (i.e. R1). Also, address the few editorial comments made in the ´Aditional Editor Comments´ space right after the signature field at the bottom of this message.

We look forward to receiving your revised manuscript.

Kind regards,

Albert Schriefer, M.D., Ph.D.

Academic Editor

PLOS ONE

Additional Editor Comments (if provided):

Introduction, line 43. I suggest adjusting ´there is a high correlation between the life cycle of the VL disease and the environmental factors involved in it ...´ to  ´there is a high correlation between the occurrence of the VL disease and the environmental factors involved in it ...´.Methods, lines 111-114. Please, refer to table 1 here: ´A total of 19 climate variables were downloaded from the 111 Worldclime website (www.worldclime.com) with a resolution of 30 sec (�1 sq.km) and were used 112 along with an elevation variable at the same resolution to evaluate and determine the appropriate 113 ecological niches (Table 1)´.Tables 2 and 3. Please, invert the tables. Titles are correct, but the corresponding tables have been inverted.Table ´The results of T-test and Kruskal –walls test for incidence of VL ...´. In the cell ´<1000 -1150>´, I suggest substituting this label with either ´Lower than 1000 and higher than 1150´, or ´<1000 and >1150´.Results, lines 175-176. Please, further clarify the sentence ´ the results of the investigation of the impact of all variables together on the model were precipitation seasonality (35.40%) and mean diurnal range (21.60%)´. As currently written it is confusing.Results, line 176. Please, correct ´(Table 1and 3)´ to (Table 1)´.

Reviewers' comments:

Reviewer's Responses to Questions

**Comments to the Author**

1. If the authors have adequately addressed your comments raised in a previous round of review and you feel that this manuscript is now acceptable for publication, you may indicate that here to bypass the “Comments to the Author” section, enter your conflict of interest statement in the “Confidential to Editor” section, and submit your "Accept" recommendation.

Reviewer #1: (No Response)

Reviewer #2: All comments have been addressed

2. Is the manuscript technically sound, and do the data support the conclusions?

Reviewer #1: Partly

Reviewer #2: Partly

3. Has the statistical analysis been performed appropriately and rigorously? 

Reviewer #1: Yes

Reviewer #2: No

4. Have the authors made all data underlying the findings in their manuscript fully available?

Reviewer #1: Yes

Reviewer #2: No

5. Is the manuscript presented in an intelligible fashion and written in standard English?

Reviewer #1: Yes

Reviewer #2: No

6. Review Comments to the Author

Reviewer #1: Dear Authors

Thank you for addressing most of the comments. Still I think two comments are not addressed:

1- Did you used the points of VL disease for Moran's analysis or considered the counties as the analysis units?

2- Fig 5 for the result of spatial auto-correlation should be replaced with a map resulted from this analysis. You can ask Arcmap to plot the map i this analysis.

Good luck

Reviewer #2: (No Response)

7. PLOS authors have the option to publish the peer review history of their article (what does this mean?). If published, this will include your full peer review and any attached files.

Reviewer #1: No

Reviewer #2: No

---

## [Author Response · Author response to Decision Letter 1]

1 Jul 2020

My dear editor,

Thanks for you and Reviewers and good time.

Additional Editor Comments (if provided):

1. Introduction, line 43. I suggest adjusting ´there is a high correlation between the life cycle of the VL disease and the environmental factors involved in it ...´ to ´there is a high correlation between the occurrence of the VL disease and the environmental factors involved in it ...´. Ok, was done. 

2. Methods, lines 111-114. Please, refer to table 1 here: ´A total of 19 climate variables were downloaded from the 111 Worldclime website (www.worldclime.com) with a resolution of 30 sec (�1 sq.km) and were used 112 along with an elevation variable at the same resolution to evaluate and determine the appropriate 113 ecological niches (Table 1)´. Ok, was done.

3. Tables 2 and 3. Please, invert the tables. Titles are correct, but the corresponding tables have been inverted. Ok, was done.

4. Table ´The results of T-test and Kruskal –walls test for incidence of VL ...´. In the cell ´<1000 -1150>´, I suggest substituting this label with either ´Lower than 1000 and higher than 1150´, or ´<1000 and >1150´. Ok, was done.

5. Results, lines 175-176. Please, further clarify the sentence ´ the results of the investigation of the impact of all variables together on the model were precipitation seasonality (35.40%) and mean diurnal range (21.60%)´. As currently written it is confusing. Ok, was done and edited (Once, the impact of each variable on the model was evaluated separately (Fig 8). In addition, the impact of all variables together on the model were precipitated again. The model showed that two factors; seasonality (35.40%) and mean diurnal range (21.60%) had the greatest impact on the occurrence of the disease)

6. Results, line 176. Please, correct ´(Table 1and 3)´ to (Table 1)´. Ok, was done.

Reviewer #1: Dear Authors

Thank you for addressing most of the comments. Still I think two comments are not addressed:

1- Did you used the points of VL disease for Moran's analysis or considered the counties as the analysis units? We used the points of VL disease for Moran's analysis.

2- Fig 5 for the result of spatial auto-correlation should be replaced with a map resulted from this analysis. You can ask Arcmap to plot the map i this analysis. To determine the autocorrelation, the Moran and G index have been used, and the Arcmap software output is in the form of a graph, not a map.

Best regards 

Dr.Eslam Moradi-Asl

---

## [Editor Report · Decision Letter 2]

8 Jul 2020

Spatial Autocorrelation and Epidemiological Survey of Visceral Leishmaniasis in an endemic area of Azerbaijan Region, Northwest of Iran

PONE-D-19-35837R2

Dear Dr. Moradi-Asl,

We’re pleased to inform you that your manuscript has been judged scientifically suitable for publication and will be formally accepted for publication once it meets all outstanding technical requirements.

Kind regards,

Albert Schriefer, M.D., Ph.D.

Academic Editor

PLOS ONE
---

## [Editor Report · Acceptance letter]

5 Aug 2020

PONE-D-19-35837R2 

Spatial autocorrelation and epidemiological survey of visceral leishmaniasis in an endemic area of Azerbaijan region, the northwest of Iran 

Dear Dr. Moradi-Asl:

I'm pleased to inform you that your manuscript has been deemed suitable for publication in PLOS ONE. Congratulations! Your manuscript is now with our production department. 

Kind regards, 

on behalf of

Dr. Albert Schriefer 

Academic Editor

PLOS ONE